# Algorithms for the preordering problem and their application to the task of jointly clustering and ordering the accounts of a social network

**Jannik Irmai**                                                  *jannik.irmai@tu-dresden.de* ⬤
*TU Dresden, Dept. of Computer Science, Dresden, Germany*

**Maximilian Moeller**                                     *maximilian.moeller@kuleuven.be* ⬤
*KU Leuven, Dept. of Computer Science, Leuven, Belgium*

**Bjoern Andres**                                               *bjoern.andres@tu-dresden.de* ⬤
*TU Dresden, Dept. of Computer Science, Dresden, Germany*
*Center for Scalable Data Analytics and AI, Dresden/Leipzig*

**Reviewed on OpenReview:** *https://openreview.net/forum?id=cBsUnv7Cb3*

## Abstract

The NP-hard maximum value preordering problem is both a joint relaxation and a hybrid of the clique partition problem (a clustering problem) and the partial ordering problem. Toward approximate solutions and lower bounds, we introduce a linear-time 4-approximation algorithm that constructs a maximum dicut of a subgraph and define local search heuristics. Toward upper bounds, we tighten a linear program relaxation by the class of odd closed walk inequalities that define facets, as we show, of the preorder polytope. We contribute implementations[1] of the algorithms, apply these to the task of jointly clustering and partially ordering the accounts of published social networks, and compare the output and efficiency qualitatively and quantitatively.

## 1 Introduction

We set out to find a binary relation on the set of accounts of a social network. Here, the notion that one account *follows, likes* or *reacts to* another account need neither be symmetric nor asymmetric. In particular, $i$ following $j$ does not need to imply that $j$ follows $i$, nor does it necessarily imply that $j$ does not follow $i$. Clustering does not capture the asymmetric subsets of the relation on the social network because the equivalence relation that characterizes the clustering is symmetric. Similarly, partial ordering does not capture the symmetric subset of the relation because partial orders are antisymmetric. Like in clustering and partial ordering, we work with the assumption[2] that the relation on the social network is transitive, i.e., if $i$ follows $j$ and $j$ follows $k$ then $i$ follows $k$. Unlike in clustering and partial ordering, we do not assume symmetry or antisymmetry. To model our assumption, we apply the preordering problem:

**Definition 1.1.** Wakabayashi (1998) Given a finite set $V$ and a (value) $c_{ij} \in \mathbb{R}$ for every pair $ij$ of distinct $i, j \in V$, the *(maximum value) preordering problem* consists in finding a preorder $\lesssim$ on $V$ so as to maximize the sum of the (values) $c_{ij}$ of those pairs $ij \in V^2$ with $i \neq j$ for which $i \lesssim j$.

If a preorder is symmetric, it is an equivalence relation and therefore characterizes a clustering. If a preorder is antisymmetric, it is a partial order. In fact, the preordering problem is a joint relaxation of the clique partition problem (a clustering problem) and the partial ordering problem. Moreover, every preorder defines, and is characterized by, a clustering and a partial order: A clustering is defined by the symmetric subset

---

[1]Source code available at: https://github.com/JannikIrmai/preordering-problem
[2]This assumption simplifies and possibly contradicts reality, and we quantify the deviation empirically.

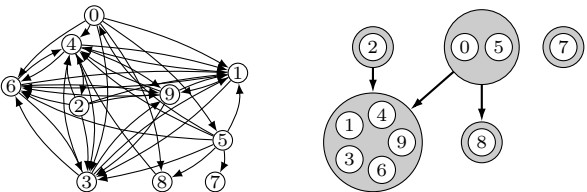

Figure 1: Depicted on the left is a Twitter ego-network from Leskovec & Mcauley (2012). Depicted on the right is a preorder on the accounts of this network consisting of a clustering (gray circles) and a partial order on the set of clusters (black edges). This preorder is an optimal solution to the instance of the preordering problem (Definition 1.1) defined such that $c_{ij} = 1$ if the pair $ij$ exists in the network, and $c_{ij} = -1$ otherwise.

of the preorder (an equivalence relation). A strict partial order *on the set of clusters* is well-defined by the asymmetric subset of the preorder. (An example is depicted in Figure 1.) Thus, the preordering problem is also a hybrid of a clustering problem and a partial ordering problem.

In this article, we define algorithms for the preordering problem. To obtain approximate solutions and lower bounds, we introduce a linear-time 4-approximation algorithm and local search heuristics. To obtain upper bounds, we consider a linear program (LP) relaxation that we tighten by the class of odd closed walk inequalities that define facets, as we show, of the preorder polytope. We contribute implementations[2] of the algorithms we discuss, apply these to social networks from Fink et al. (2023) and Leskovec & Mcauley (2012) with up to $10^7$ edges and compare the output and efficiency qualitatively and quantitatively.

## 2 Related work

The preordering problem is formulated by Wakabayashi (1998) who establishes NP-hardness even for values in $\mathcal{O}(|V|^6)$ and discusses the complexity of many specializations. Her hardness result for the general problem is strengthened by Weller et al. (2012) who prove NP-hardness even for values in $\{-1, 1\}$. A local search algorithm that greedily decides whether or not to relate pairs is introduced by Böcker et al. (2009). We reproduce this algorithm in Section 4.2. The geometry of the preorder polytope is discussed briefly by Gurgel (1992).

Closely related to preordering but studied independently in the literature is the transitivity editing problem Jacob et al. (2008); Böcker et al. (2009); Weller et al. (2012). Given a digraph, the objective is to find a minimum set of arcs to be added or removed so as to make the digraph transitive. This problem is equivalent to the preordering problem with values $c_{ij} = 1$ if $ij$ is an arc in the digraph, and $c_{ij} = -1$ otherwise. Even this restriction to values in $\{-1, 1\}$ is NP-hard Weller et al. (2012). The transitivity editing problem is studied predominantly from the perspective of fixed-parameter tractability; efficient algorithms for the case where the total number of arcs to be added or removed is bounded are developed by Böcker et al. (2009); Weller et al. (2012).

Preordering is related to correlation clustering Emanuel & Fiat (2003); Bansal et al. (2004); Demaine et al. (2006), more specifically, to maximum value weighted correlation clustering for complete graphs, which is also known as clique partitioning Grötschel & Wakabayashi (1989), where values $c_{ij}$ are associated with unordered pairs $\{i, j\}$, and the task is to find an equivalence relation $\sim$ on $V$ so as to maximize the sum of the values of those pairs $\{i, j\}$ for which $i \sim j$. This problem is more specific than preordering in that equivalence relations are precisely the symmetric preorders. The complexity and hardness of approximation of correlation clustering have been studied extensively (Swamy, 2004; Charikar et al., 2005; Chawla et al., 2015; Veldt, 2022; Cohen-Addad et al., 2022; Klein et al., 2023). In particular, it is known that clique partitioning cannot be approximated to within $n^{1-\epsilon}$ unless P = NP for $n$ the number of elements and any $\epsilon > 0$ (Bachrach et al., 2013; Zuckerman, 2007). Much work has been devoted also to the study of the clique partitioning polytope Grötschel & Wakabayashi (1990b;a); Deza et al. (1991; 1992); Chopra & Rao (1995); Bandelt et al. (1999); Oosten et al. (2001); Sørensen (2002); Letchford & Sørensen (2025). Correlation clustering has many applications, including community detection in social networks (Brandes et al., 2008;

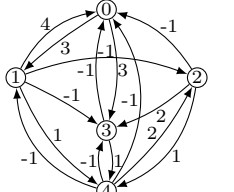 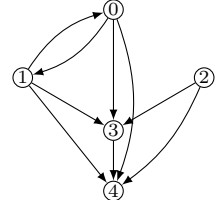 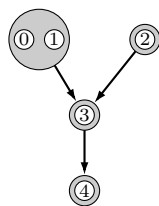

Figure 2: Depicted on the left is an instance of the preordering problem with $|V| = 5$. The value $c_{ij}$ is written next to the arc from $i$ to $j$, and arcs that are not depicted have a value of 0. Depicted in the middle is an optimal solution to the preordering problem. The value of this solution is $P(c) = 14$. The upper bound is $B(c) = 17$ and thus, the transitivity index is $T(c) = \frac{14}{17}$. Depicted on the right is the same preorder as a partial order on clusters (equivalence classes). Here and in all subsequent figures, partial orders are depicted by their unique transitive reduction. E.g., the arc $(2, 4)$ is not shown because it is implied by the arcs $(2, 3)$ and $(3, 4)$.

Veldt et al., 2018) and image analysis (Yarkony et al., 2012; Beier et al., 2015; Aflalo et al., 2023; Abbas & Swoboda, 2023). Variants that emphasize local errors are studied by Puleo & Milenkovic (2016); Kalhan et al. (2019); Ahmadian et al. (2020).

Preordering is related also to partial ordering (Müller, 1996) where the task is to find a partial order $\leq$ on $V$ so as to maximize the sum of the values of those pairs $ij$ for which $i \leq j$. Partial ordering is more specific than preordering in that partial orders are precisely the antisymmetric preorders. The polyhedral geometry of partial orders is studied by Müller (1996). Also this problem has received less attention so far than even more constrained problems, like the linear ordering problem (Grötschel et al., 1984; Martí & Reinelt, 2011; Ceberio et al., 2015), the closely related rank aggregation problem (Ailon et al., 2008; Schalekamp & Zuylen, 2009) and the feedback arc set problem on tournament graphs (Karpinski & Schudy, 2010). The geometry of several order polytopes is studied by Doignon & Fiorini (2001); Doignon & Rexhep (2016). The understanding of the computational complexity of finding a maximum value order for various types of orders is improved by Hudry (2008; 2012) who shows that some problems remain NP-hard for restricted value functions.

A joint relaxation of correlation clustering and *linear* ordering is the bucket ordering problem, sometimes called the weak ordering problem, that asks for both a clustering and a *linear* order on the clusters (Gurgel & Wakabayashi, 1997; Fiorini, 2003; Fiorini & Fishburn, 2004; Fiorini, 2006). This problem has been applied to the tasks of dating archeological sites (Gionis et al., 2006), finding consensus among voters Aledo et al. (2018; 2021) and constructing user recommendations (Jurewicz et al., 2023).

## 3 Preordering problem

Throughout this article, consider some finite, non-empty set $V$, e.g., the accounts of a social network. For simplicity and conciseness, let $V_n$ denote the set of all $n$-tuples of pairwise distinct elements of $V$. Refer to the elements of $V$ and $V_2$ as *nodes* and *arcs*, respectively.

We cast the preordering problem (Definition 1.1) with respect to $V$ and values $c \colon V_2 \to \mathbb{R}$ as an integer linear program (ILP):

$$\max \quad \sum_{ij \in V_2} c_{ij} x_{ij} \tag{1}$$

$$\text{subject to} \quad x_{ij} \in \{0, 1\} \qquad \forall\, ij \in V_2 \tag{2}$$

$$x_{ij} + x_{jk} - x_{ik} \leq 1 \qquad \forall\, ijk \in V_3 \tag{3}$$

An example is depicted in Figure 2. Let $P(c)$ denote the objective value of an optimal solution. For convenience, define $x_{ii} = 1$ and $c_{ii} = 0$ for all $i \in V$. Note that the map from $x$ to $\lesssim_x := \{ij \in V^2 \mid x_{ij} = 1\}$ is a bijection from the feasible solutions of the ILP (1)–(3) to the set of preorders on $V$. Indeed, the objective

value of a feasible solution $x$ to the ILP (1)–(3) is equal to the value of $\lesssim_x$ according to Definition 1.1. See Appendix A.1 for further details.

**Theorem 3.1** (Weller et al. (2012))**.** *The preordering problem is* NP*-hard even for* $c\colon V_2 \to \{-1, 1\}$*.*

In the context of clustering, it is well-known that clique partitions are precisely the complements of unions of cuts. The following lemma states analogously that preorders are precisely the complements of unions of directed cuts (dicuts):

**Lemma 3.2.** *An arc subset $A \subseteq V^2$ is a preorder if and only if $V^2 \setminus A$ is the union of dicuts, i.e., $V^2 \setminus A = \bigcup_{S \in \mathcal{S}} \delta(S)$ for some $\mathcal{S} \subseteq 2^V$ where $\delta(S) = \{ij \in V_2 \mid i \in S, j \notin S\}$ denotes the dicut in $V_2$ defined by $S$.*

*Proof. If.* Let $V^2 \setminus A = \bigcup_{S \in \mathcal{S}} \delta(S)$ for some $\mathcal{S} \subseteq 2^V$. Suppose $A$ is not a preorder, i.e., there exist $i, j, k \in V$ such that $ij, jk \in A$ but $ik \notin A$. By definition of $A$, there exists $S \in \mathcal{S}$ such that $ij, jk \notin \delta(S)$ and $ik \in \delta(S)$. By definition of directed cuts, $ik \in \delta(S)$ implies $i \in S$ and $k \notin S$. With this, $ij \notin \delta(S)$ implies $j \in S$ while $jk \notin \delta(S)$ implies $j \notin S$, a contradiction.

*Only if.* Let $A \subseteq V^2$ be a preorder. For $i \in V$, define $S_i := \{j \in V \mid ij \in A\}$ and let $\mathcal{S} = \{S_i \subseteq V \mid i \in V\}$. We show that $V^2 \setminus A = \bigcup_{S \in \mathcal{S}} \delta(S)$ holds. Firstly, suppose $ij \in \bigcup_{S \in \mathcal{S}} \delta(S)$, i.e., there exists $k \in V$ such that $ij \in \delta(S_k)$. By definition this implies $ki \in A$ and $kj \notin A$. As $A$ is a preorder it follows that $ij \notin A$. Secondly, suppose $ij \in A$. For any $k \in V$ it holds that $i \in S_k$ iff $ki \in A$ iff $kj \in A$ iff $j \in S_k$. Therefore $ij \notin \delta(S_k)$ for all $k \in V$ and in particular $ij \notin \bigcup_{S \in \mathcal{S}} \delta(S)$, which proves the claim. $\square$

Next, we introduce a notion to quantify the transitivity of a value function $c$. Clearly, the sum of all positive values is an upper bound to the preordering problem. Let

$$T(c) = \frac{P(c)}{B(c)} \quad \text{with} \quad B(c) = \sum_{ij \in V_2 \,:\, c_{ij} > 0} c_{ij} \tag{4}$$

denote the proportion of the optimal value to the upper bound. We call it the *transitivity index*. It quantifies how transitive the values $c$ are. If there exists a transitive relation that contains all pairs with positive value and no pair with negative value, then $P(c) = B(c)$, hence $T(c) = 1$. If no such relation exists, then $T(c) < 1$. As the empty relation is transitive, $P(c) \geq 0$, and thus, $T(c) \in [0, 1]$ for all $c\colon V_2 \to \mathbb{R}$. In the context of social networks, the transitivity index can be interpreted as a measure for how homogeneous the information flow is. If the transitivity index is high, the information flow is homogeneous: If $a$ receives information from $b$ and $b$ receives information from $c$, the $a$ also receives information form $c$. If the index is low, the flow of information is less structured.

## 4 Algorithms

In this section, we describe algorithms for solving the preordering problem and for efficiently computing upper bounds. The input size of an instance of the preordering problem is $|V_2| \in \mathcal{O}(|V|^2)$. Nevertheless, we may state the time complexity of an algorithm as a function of $|V|$.

### 4.1 Linear-time 4-approximation via max-dicut

We exploit a connection between directed cuts and preorders to obtain a 4-approximation algorithm for the preordering problem. For a directed graph (digraph) $D = (V, E)$ and a node subset $S \subseteq V$, the *directed cut (dicut)* defined by $S$ is the edge subset $\delta(S) = \{ij \in E \mid i \in S, j \notin S\}$. Notice that any dicut is transitive because it does not contain any consecutive edges. Given edge weights $w\colon E \to \mathbb{R}_{\geq 0}$, the *maximum directed cut problem (max-dicut)* consists in finding a dicut of maximum weight. Max-dicut admits a simple randomized 4-approximation: By assigning each node with probability $\frac{1}{2}$ to $S$, the probability that any given edge $ij \in E$ is contained in the cut is $\frac{1}{4}$ (namely if $i \in S$ and $j \in S$, each with probability $\frac{1}{2}$). Therefore, the expected value of such a randomly chosen dicut is $\frac{1}{4} \sum_{ij \in E} w_{ij}$ where $\sum_{ij \in E} w_{ij}$ is a trivial upper bound to the max-dicut problem. This algorithm can be de-randomized (Halperin & Zwick, 2001; Bar-Noy & Lampis, 2012) as detailed in Appendix A.2 to obtain Algorithm 1 that runs in time $\mathcal{O}(|V|^2)$.

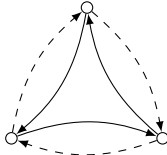

Figure 3: Depicted is an instance of the preordering problem with $|V| = 3$. Solid and dashed arcs indicate values of $+1$ and $-1$, respectively. Here, $P(c) = 1$, $B(c) = 3$ and $T(c) = \frac{1}{3}$.

**Theorem 4.1.** *There exists a 4-approximation algorithm for the preordering problem with time complexity* $\mathcal{O}(|V|^2)$.

*Proof.* For any instance of the preordering problem given by $V$ and $c \colon V_2 \to \mathbb{R}$, let $D = (V, E)$ with $E = \{ij \in V_2 \mid c_{ij} > 0\}$ and edge weights $w_{ij} = c_{ij}$ for all $ij \in E$ be the digraph that consists of all arcs with positive value. Algorithm 1 finds a dicut $\delta(S)$ with value $\sum_{ij \in \delta(S)} w_{ij} \geq \frac{1}{4} \sum_{ij \in E} w_{ij}$. As $\delta(S)$ does not contain any two consecutive edges, $x \in \{0, 1\}^{V_2}$ with $x_{ij} = 1$ if and only if $ij \in \delta(S)$ is feasible for the instance of the preordering problem. Also,

$$\sum_{ij \in V_2} c_{ij} x_{ij} = \sum_{ij \in \delta(S)} w_{ij} \geq \frac{1}{4} \sum_{ij \in E} w_{ij} = \frac{1}{4} B(c) \ ,$$

i.e., $x$ is a 4-approximate solution. The computations described above and Algorithm 1 are executed in time $\mathcal{O}(|V|^2)$. $\qquad\square$

*Remark* 4.2. An immediate consequence of the proof of Theorem 4.1 is that $T(c) \geq \frac{1}{4}$ holds for all $c \colon V_2 \to \mathbb{R}$. By the example in Figure 3, there exists a $c \colon V_2 \to \mathbb{R}$ with $T(c) = \frac{1}{3}$. We do not know whether there exist $c \colon V_2 \to \mathbb{R}$ with $T(c) < \frac{1}{3}$.

*Remark* 4.3. Every dicut is also a partial order and, thus, the partial ordering problem also admits a 4-approximation.

### 4.2 Greedy arc fixation (Böcker et al., 2009)

Böcker et al. (2009) propose an algorithm that fixes arc labels greedily, one arc at a time. The decision which arc to fix and to what label is based on the induced cost of these decisions: Excluding an arc $ij$ from the preorder (i.e., setting $x_{ij} = 0$) implies that for every $k \in V \setminus \{i, j\}$ at most one of the arcs $ik$, $kj$ can be included in the preorder. If both these arcs have a positive value, at least the smaller of the two values

---

**Algorithm 1:** Greedy max-dicut approximation

**Input:** Digraph $D = (V, E)$, weights $w \colon E \to \mathbb{R}_{\geq 0}$
$S := \emptyset, \ \bar{S} := \emptyset$
$g_i := \sum_{j \in V : ij \in E} w_{ij} - \sum_{j \in V : ji \in E} w_{ji} \quad \forall i \in V$
**while** $S \cup \bar{S} \neq V$
    $i \in \underset{j \in V \setminus (S \cup \bar{S})}{\operatorname{argmax}} |g_i|$
    **if** $g_i \geq 0$
        $S := S \cup \{i\}$
        $g_j := g_j - w_{ij} \quad \forall \, j \in V : ij \in E$
        $g_j := g_j - w_{ji} \quad \forall \, j \in V : ji \in E$
    **else**
        $\bar{S} := \bar{S} \cup \{i\}$
        $g_j := g_j + w_{ij} \quad \forall \, j \in V : ij \in E$
        $g_j := g_j + w_{ji} \quad \forall \, j \in V : ji \in E$
**return** $\delta(S)$

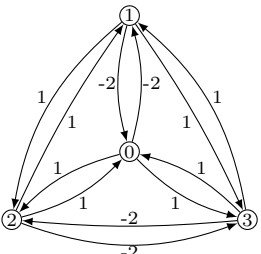

Figure 4: For the instance of the preordering problem depicted above the optimal objective value is 4. The greedy arc fixation algorithm may, in this order, fix 01, 10, 12, 02, 03, 13 to 1 and then fix the remaining arcs to 0 yielding a solution with objective value 0. We note that this solution is obtained if the algorithm breaks ties unfavorably. However, there exist more complex instances where the algorithm still fails to come within a factor of 4 of the optimal objective value (cf. Appendix A.3).

is "lost" by excluding $ij$ from the preorder. In total, the induced cost for excluding $ij$ from the preorder is computed as

$$\text{ice}(ij) = \max(0, c_{ij}) + \sum_{k \in V \setminus \{i,j\}} \max\{0, \min\{c_{ik}, c_{ki}\}\} \ .$$

Similarly, the induced cost of inserting an arc $ij$ into the preorder (i.e., setting $x_{ij} = 1$) is given by

$$\text{ici}(ij) = \max(0, -c_{ij}) + \sum_{k \in V \setminus \{i,j\}} (\max\{0, \min\{c_{ki}, -c_{kj}\}\} + \max\{0, \min\{c_{jk}, -c_{ki}\}\}) \ .$$

The greedy algorithm by Böcker et al. (2009) starts with all arcs being unlabeled and iteratively selects an unlabeled arc $ij$ for which $\max\{\text{ice}(ij), \text{ici}(ij)\}$ is maximal. It labels $x_{ij} = 0$ if $\text{ice}(ij) \geq \text{ici}(ij)$, and $x_{ij} = 1$ otherwise. After labeling a arc $ij$, its value $c_{ij}$ is set to $+\infty$ if $x_{ij} = 1$ and $-\infty$ otherwise. This ensures convergence to a feasible solution. We term this algorithm *greedy arc fixation*. We note that Böcker et al. (2009) suggest to include a lower bound on the disagreement on the instance excluding the nodes $i$ and $j$ in the computation of the induced costs ice and ici. We do not use such additional bounds as these have led to worse solutions and longer computation times in our experiments.

In total, the greedy arc fixation algorithm consists of $|V_2|$ iterations. By maintaining a priority queue sorted by the induced costs, an arc with maximal induced cost can be queried in time $\mathcal{O}(\log |V|)$. After fixing an arc $ij$, only the induced costs of the arcs incident to $i$ and $j$ change. For a single arc, this change can be computed in constant time. And the changed induced cost can be updated in the priority queue in time $\mathcal{O}(\log(|V|))$. This results in a time of $\mathcal{O}(|V| \log |V|)$ per iteration and in a total runtime of $\mathcal{O}(|V|^3 \log |V|)$.

While the greedy arc fixation algorithm performs well in practice (see Section 5), it does not guarantee any approximation factor. This is shown by the example in Figure 4 that illustrates that the solution computed by the greedy arc fixation algorithm can be off by an arbitrary factor.

### 4.3 Greedy arc insertion

Next, we define a greedy algorithm for the preordering problem that starts from any feasible solution (e.g., the identity relation) and greedily inserts pairs into the relation without violating transitivity. While conceptually similar to greedy additive edge contraction (Keuper et al., 2015) and agglomerative clustering (Bailoni et al., 2022), specific combinatorial challenges arise from the preordering problem: Firstly, there are up to $\mathcal{O}(|V|^2)$ edge insertions in preordering, in contrast to at most $\mathcal{O}(|V|)$ edge contractions in clustering (because the number of clusters decreases by one in every iteration). Secondly, in each iteration, $\mathcal{O}(|V|^2)$ arcs need to be considered to potentially be added to the relation. Adding any one arc can result in $\mathcal{O}(|V|^2)$ additional arcs that need to be added to satisfy transitivity. A naïve implementation of this algorithm would require $\mathcal{O}(|V|^4)$ operations per iteration. Below, we describe a more efficient implementation: Suppose $x \colon V_2 \to \{0, 1\}$ is a

feasible solution to the preordering problem, and let $ij \in V_2$. By setting $x_{ij} = 1$, all arcs $k\ell$ with $x_{ki} = x_{j\ell} = 1$ need to be set to 1 as well in order to satisfy transitivity. The amount by which the total value changes can thus be calculated as

$$g_{ij} = \sum_{k,\ell \in V} c_{k\ell} \left(1 - x_{k\ell}\right) x_{ki} \, x_{j\ell} \tag{5}$$

where $c_{k\ell} \left(1 - x_{k\ell}\right)$ ensures that only values of arcs that are currently labeled 0 are considered. We call this value the *gain* from setting $ij$ to 1. By representing $x$, $c$, and $g$ as matrices, the gain matrix $g$ can be calculated by two matrix multiplications

$$g = x^\top (c \odot (1 - x)) x^\top \tag{6}$$

where $\odot$ denotes element-wise multiplication. With this, each iteration can be performed with only $\mathcal{O}(|V|^3)$ operations. Furthermore, the matrix multiplication can be carried out in parallel on a GPU. In our implementation of this algorithm, we do not utilize GPU parallelization but instead use a sequential method that exploits the sparsity of the 01-matrix $x$.

## 4.4 Greedy moving

The greedy arc insertion algorithm never removes an arc from a feasible solution. Removing an arc from a preorder can result in a relation that is not a preorder and thus not a feasible solution. Removing a dicut from a preorder results in another preorder (by Lemma 3.2) and thus maintains feasibility. However, finding a dicut of minimum cost (in order to maximize the objective of the solution obtained by removing the cut) is well-known to be NP-hard (Karp, 1972). Below, we define $O(|V|^2)$ operations that may remove arcs from the preorder such that the difference in value resulting from each operation can be calculated efficiently.

- *Move up (down)*: Remove a node $i \in V$ from its current equivalence class and place it immediate above (below) the equivalence class in the partial order. That is, set $x_{ji} = 0$ ($x_{ij} = 0$) for all $j \in V \setminus \{i\}$ with $x_{ij} = x_{ji} = 1$.

- *Move equivalence class*: Move a node $i \in V$ from its current equivalence class to the equivalence class of node $j \in V \setminus \{i\}$. That is, set $x_{ik} = x_{jk}$, $x_{ki} = x_{ki}$ for all $k \in V \setminus \{i\}$.

- *Remove arcs from transitive reduction*: Remove an arc from the transitive reduction of the partial order of equivalence classes. That is, for an arc $ij \in V_2$ such that $x_{ik} = 0 \lor x_{kj} = 0$ for all $k \in V \setminus \{i,j\}$, set $x_{i'j'} = 0$ for all $i' \in V$ with $x_{ii'} = x_{i'i} = 0$ and all $j' \in V$ with $x_{jj'} = x_{j'j} = 1$.

We implement an algorithm, termed *greedy moving*, that, in each iteration, performs an optimal one of the moves described above, or an arc insertion according to section 4.3. The operation that increases the objective value maximally can be found in $\mathcal{O}(|V|^3)$, similarly to section 4.3.

## 4.5 Linear programming

In order to obtain an upper bound on the optimal value of the preordering problem and thus an upper bound on the transitivity index $T$ from (4), we consider the LP relaxation fo the ILP (1)–(3) obtained by replacing (2) with

$$x_{ij} \in [0,1] \qquad\qquad \forall \, ij \in V_2 \; . \tag{7}$$

This bound can be strengthened by adding linear inequalities to the formulation that are satisfied by all binary solutions and cut off fractional solutions. Particularly powerful are inequalities that induce facets of the *preorder polytope*, i.e., the convex hull of the feasible solutions to the preordering problem. Polytopes of several closely related problems have been studied extensively, as discussed in Section 2. Müller (1996) introduce the class of *odd closed walk inequalities* for the partial order polytope. We adapt these to the preorder polytope as follows.

**Definition 4.4.** Let $k \geq 3$ odd and $v\colon \{0, \dots, k-1\} \to V$. The *odd closed walk inequality* with respect to $v$ is defined below where the additions $i+1$ and $i+2$ are modulo $k$.

$$\sum_{i=0}^{k-1} \left( x_{v_i v_{i+1}} - x_{v_i v_{i+2}} \right) \leq \frac{k-1}{2} \tag{8}$$

Müller (1996) proves validity of the odd closed walk inequalities for the partial order polytope by showing that they are obtained by adding certain triangle inequalities (3) and box inequalities $0 \leq x_{ij}$ for several $ij \in V_2$ and rounding down the right-hand side. As triangle and box inequalities are valid also for the preorder polytope, the odd closed walk inequalities are valid for the preorder polytope as well. Moreover:

**Lemma 4.5.** *Every inequality that induces a facet of the partial order polytope and is valid for the preorder polytope also induces a facet of the preorder polytope.*

*Proof.* As every partial order is a preorder, the partial order polytope is contained in the preorder polytope. Thus, for any inequality that is valid for both polytopes, the dimension of the induced face of the preorder polytope is greater than or equal to the dimension of the induced face of the partial order polytope. Since the partial order polytope is full-dimensional (Müller, 1996, Theorem 3.1), so is the preorder polytope, and, in particular, they have the same dimension. Together, any inequality that induces a facet of the partial order polytope, i.e., a face of co-dimension 1, and is valid for the preorder polytope, induces a face of the preorder polytope of co-dimension at most 1, and, therefore, defines a facet. □

**Theorem 4.6.** *Let $k \geq 3$ odd and let $v\colon \{0, \dots, k-1\} \to V$ injective (i.e., $v$ defines a simple cycle). Then the odd closed walk inequality* (8) *defines a facet of the preorder polytope.*

*Proof.* Müller (1996) shows that if the odd closed walk is a cycle (i.e., every node is visited at most once) then the corresponding odd closed walk inequality defines a facet of the partial order polytope (Müller, 1996, Theorem 4.2). Together with Lemma 4.5 the theorem follows. □

While the number of odd closed walk inequalities is exponential in $|V|$, the separation problem can be solved in polynomial time (Müller, 1996). This implies that also the LP (1), (3), (8), (7) can be solved in polynomial time (Grötschel et al., 1981).

## 5 Experiments

We apply the algorithms defined in Section 4 to instances of the preordering problem from social networks. We solve LPs and ILPs with Gurobi Optimization, LLC (2024) in such a way that triangle inequalities (3) and odd closed walk inequalities (8) are not enumerated in advance but are only added to the system if they become violated throughout the execution of the solver. All experiments are performed on an Intel Core i9-12900KF Gen12 × 24 and an NVIDIA GeForce RTX 4080 Super. We abbreviate the algorithms as follows:

- ILP: Gurobi applied to the system (1)–(3)
- LP: Gurobi applied to the system (1), (3), (7)
- OCW: Gurobi applied to the system (1), (3), (7), (8)
- GDC: Greedy max-dicut approximation (Section 4.1)
- GAF: Greedy arc fixation (Section 4.2)
- GAI: Greedy arc insertion (Section 4.3)
- GM: Greedy moving (Section 4.4)
- X+Y: Algorithm Y initialized with the output of X

### 5.1 Ego networks

In this section we consider ego networks of Twitter and Google+ (Leskovec & Mcauley, 2012). The ego network corresponding to a given account consists of one node for each account that the given account

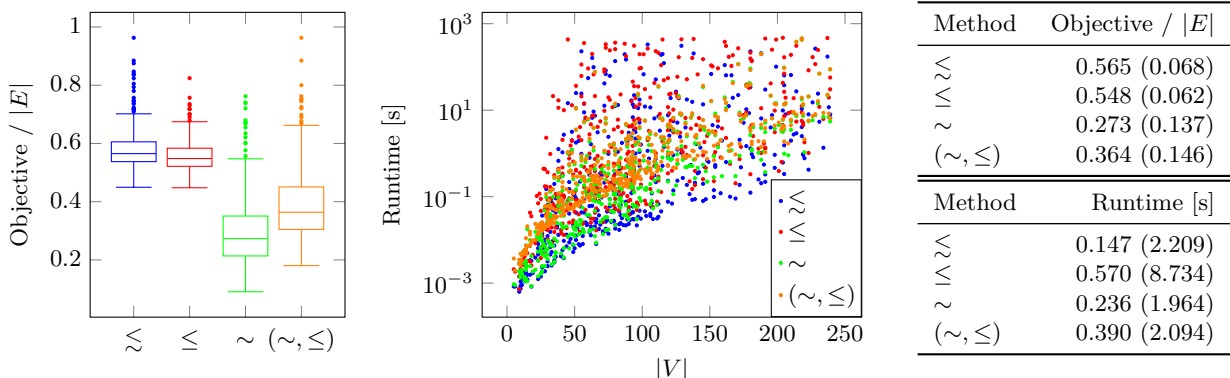

Figure 5: Shown above are objective values normalized by the number of edges (left) and runtimes (middle) for those 435 Twitter instances for which the preorder ($\lesssim$), partial order ($\leq$), and clustering ($\sim$) problems are solved to optimality within a time limit of 500 s. Here, ($\sim, \leq$) denotes successive clustering and partial ordering. Corresponding medians are reported in the tables on the right with the interquartile range in parentheses.

follows. Directed edges between accounts $i$ and $j$ indicate that $i$ follows $j$. The Twitter dataset consists of 973 ego networks with up to 247 nodes and 17930 edges. The Google+ dataset consists of 132 ego networks with up to 4938 nodes and 1614977 edges. We define instances of the preordering problem for each social network with values $c_{ij} = 1$ if the edge $ij$ is in the network and $c_{ij} = -1$ otherwise, for all $ij \in V_2$. With these values, $B(c) = |E|$.

### 5.1.1 Comparison of preordering, clustering and partial ordering

We compare solutions to the preordering problem with those obtained by solving a clustering and a partial ordering problem. Additionally, and for a comparison, we compute preorders by first computing a clustering and then computing a partial order on the obtained clusters. This successive approach is in contrast to the joint clustering and partial ordering using the preordering problem. In order to compute solutions to the clustering and partial ordering problem, we add additional inequalities to the system (1)–(3): For clustering, we add symmetry constraints $x_{ij} = x_{ji}$ for all $ij \in V_2$. For partial ordering we add antisymmetry constraints $x_{ij} + x_{ji} \leq 1$ for all $ij \in V_2$.

Results for six ego networks are shown in Figure 6 and compared quantitatively in Table 1. The first network, for instance, has $|E| = 54$ edges and admits a preorder of value 52. This preorder disagrees with two of the edges. Its transitivity index is $T(c) = \frac{52}{54} \approx 0.963$. The optimal clustering and partial ordering disagree with 14 and 20 edges, respectively. Not every social network has such a high transitivity index. The fourth network, for example, has a transitivity index of $T(c) = \frac{27}{51} \approx 0.529$, and the objective values of the optimal solutions to the clustering (20) and partial ordering (26) problems are not much smaller than the value of the optimal preorder (27).

Objective values and runtimes for the 434 out of 973 instances that have been solved within a time limit of 500 s are visualized in Figure 5. The objective values are highest for the preordering problem and lowest for the clustering problem. The values obtained by the successive approach are significantly smaller than those obtained by solving the preordering problem directly. This can be understood as an advantage of solving a joint clustering and partial ordering problem over solving these problems successively. On average, the runtime of the clustering problem is the lowest, while that of the partial ordering problem is the highest. The runtime of successive clustering and partial ordering is only marginally greater than that of just clustering because the instance of the partial ordering problem on clusters is much smaller than the instance of the partial ordering problem on individual accounts.

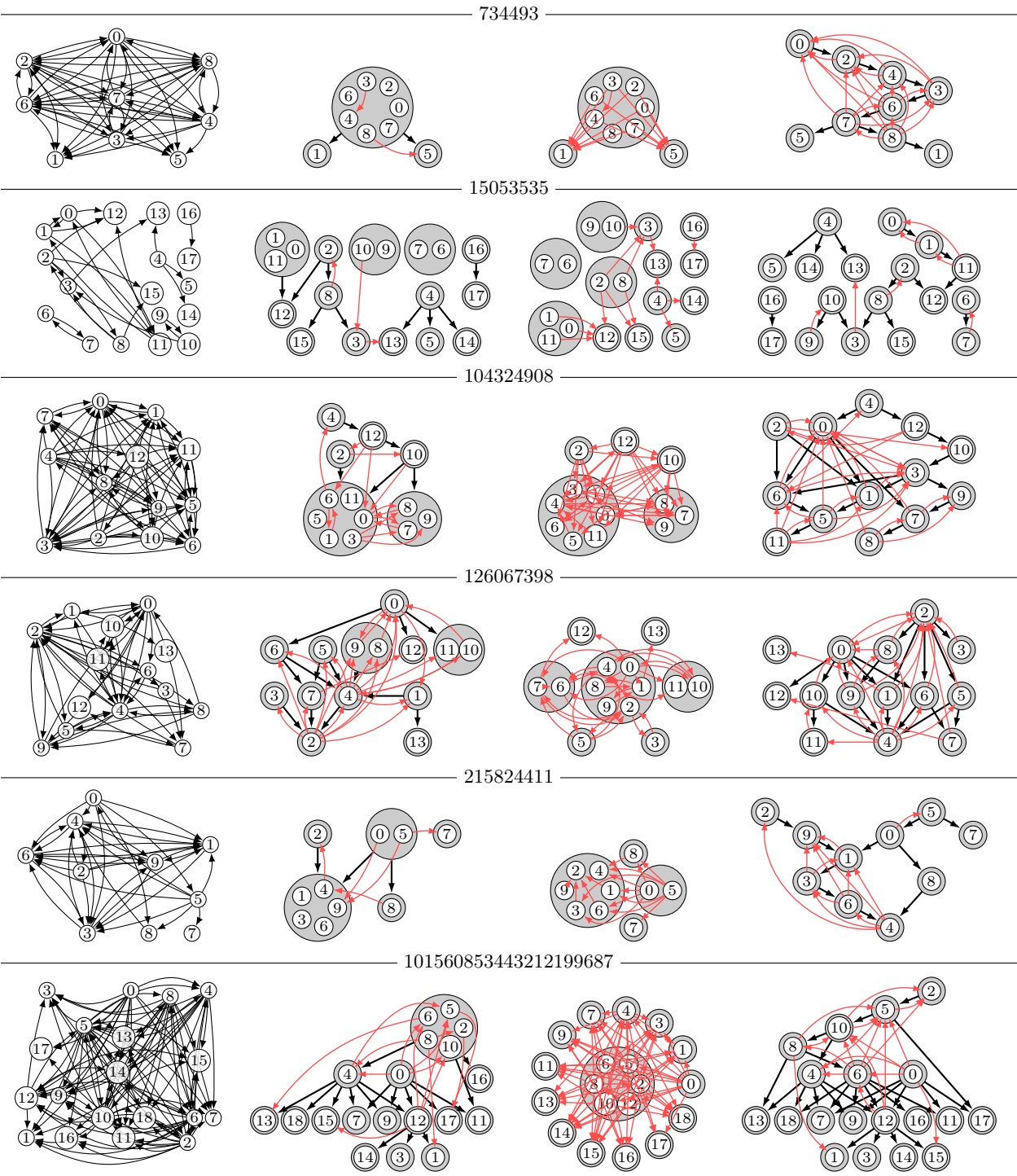

Figure 6: Depicted above in the left column are the ego networks from Table 1. Next to these networks, from left to right, are an optimal preorder, optimal clustering and optimal partial order. Disagreements with the social network are highlighted by red arcs. The total number of disagreements is precisely the number of edges minus the objective value stated in Table 1. It can be observed that the preorder has less disagreement compared to the clustering and partial order. For the network at the top, for instance, the preorder has only two disagreeing arcs while the clustering has 14 disagreeing arcs and the partial order has 20 disagreeing arcs.

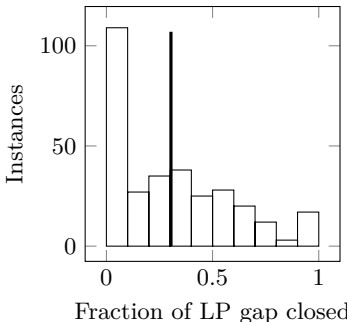
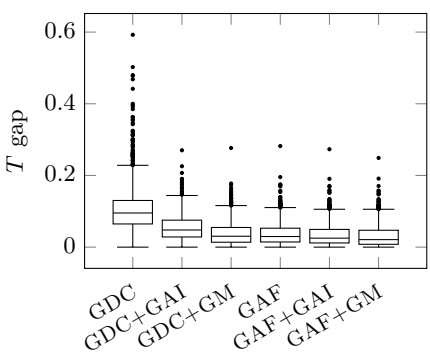

| Method | Median $T$ gap | Opt |
|---|---|---|
| GDC | 0.095 (0.065) | 2 |
| GDC+GAI | 0.047 (0.046) | 16 |
| GDC+GM | 0.030 (0.041) | 38 |
| GAF | 0.029 (0.038) | 40 |
| GAF+GAI | 0.025 (0.037) | 53 |
| GAF+GM | 0.021 (0.039) | 70 |

Figure 7: Shown on the left is the fraction of the LP gap that is closed by including odd closed walk inequalities. On average, 30.4% of the gap is closed. This evaluation is restricted to those 313 instances from the Twitter dataset for which the optimal solution is computed within a 500 s time limit and the canonical LP bound is not tight. Shown in the middle are the differences between the lower bounds computed by the heuristic algorithms and the LP bound on all 973 instances. The table on the right reports the corresponding medians differences with the interquartile range in parentheses, as well as the number of instances for which an optimal solution was found.

### 5.1.2 Optimal solutions and bounds

Within the time limit of 500 s, the ILP algorithm finds optimal solutions to the instances of the preordering problem for 499 out of 973 Twitter ego networks. For these instances, the average transitivity index $T$ is 0.580. For 186 of these instances, the LP bound according to Section 4.5 is tight. For the 313 instances for which the LP bound is not tight, the gap between the LP bound and the optimal solution is reduced by 30.4% on average by the odd closed walk inequalities; cf. Figure 7 (left).

We compute feasible solutions that are not guaranteed to be optimal with the heuristic algorithms GDC, GDC+GAI, GDC+GM, GAF, GAF+GAI, GAF+GM. The values of these feasible solutions are lower bounds on the transitivity index. Complementary to this, LP computes upper bounds on the transitivity index. Differences between these lower and upper bounds are shown in Figure 7 (right). On average, GAF+GM comes closest to the LP bound with a median difference of 0.0210. Still, for example, on 121 instances, the solution computed by GDC+GAI is strictly better than that of GAF+GM.

On those 474 instances for which the ILP could not be solved to optimality, the median gap after the time limit of 500 s is reached is 0.146. In contrast to this, the median gap between the best lower bound and the upper bound computed with OCW is 0.068. While it is NP-hard to compute the transitivity index exactly, this result shows that it can be estimated closely in polynomial time for these instances. Moreover, this shows that the discussed heuristic algorithms together with the bounds obtained by OCW can result in much better solutions than those found by a canonical ILP solver within the time limit.

Table 1: Numerical results for the six ego networks depicted in Figure 6. Reported are the number of nodes ($|V|$) and edges ($|E|$), the optimal values of the preordering ($\lesssim$), clustering ($\sim$) and partial ordering ($\leq$) problems, and the corresponding normalized objective values (note that $\lesssim/|E|$ is the transitivity index $T$).

| Platform | ID | $|V|$ | $|E|$ | $\lesssim$ | $\sim$ | $\leq$ | $\lesssim/|E|$ | $\sim/|E|$ | $\leq/|E|$ |
|---|---|---|---|---|---|---|---|---|---|
| Twitter | 734493 | 9 | 54 | 52 | 40 | 34 | 0.963 | 0.741 | 0.630 |
| Twitter | 15053535 | 18 | 26 | 23 | 12 | 19 | 0.885 | 0.462 | 0.731 |
| Twitter | 104324908 | 13 | 77 | 62 | 34 | 51 | 0.805 | 0.442 | 0.662 |
| Twitter | 126067398 | 14 | 51 | 27 | 20 | 26 | 0.529 | 0.392 | 0.510 |
| Twitter | 215824411 | 10 | 40 | 35 | 24 | 28 | 0.875 | 0.600 | 0.700 |
| Google+ | 101560853443212199687 | 19 | 107 | 92 | 24 | 87 | 0.885 | 0.224 | 0.813 |

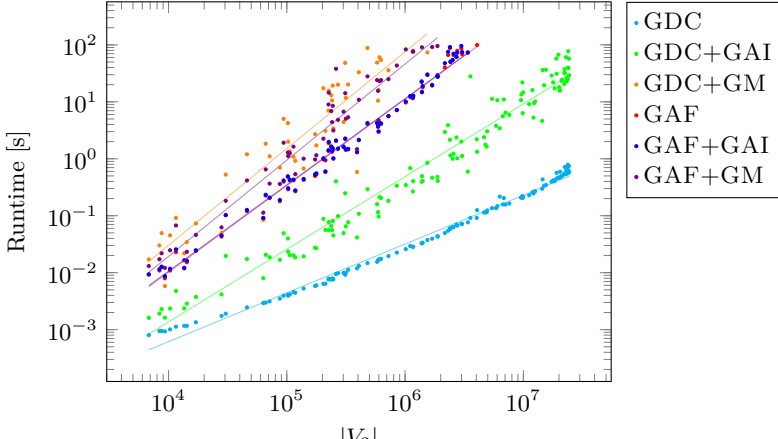

| Method | a | x |
|---|---|---|
| GDC | $2.285 \cdot 10^7$ | 0.857 |
| GDC+GAI | $1.070 \cdot 10^8$ | 1.276 |
| GDC+GM | $4.814 \cdot 10^9$ | 1.700 |
| GAF | $8.545 \cdot 10^9$ | 1.519 |
| GAF+GAI | $9.272 \cdot 10^9$ | 1.515 |
| GAF+GM | $3.401 \cdot 10^9$ | 1.689 |

Figure 8: Runtimes of six heuristic algorithms as a function of input size $|V_2|$. Results are shown for runs terminating within 100s. The fit lines are of the form $a\,|V_2|^x$ with coefficients $a$ and $x$ as stated in the table. For example, for GDC, $x$ is close to 1 which is expected as its asymptotic runtime is linear in $|V_2|$ (Theorem 4.1).

### 5.1.3 Efficiency of heursitic algorithms

We analyze the empirical runtime of the heuristic algorithms on the much larger Google+ ego networks. The largest network contains $|V| = 4,938$ nodes, which results in an instance of the preordering problem of size $|V_2| = 24,378,906$. The runtimes are shown in Figure 8. The fastest algorithm is GDC. This is expected as its runtime is linear in the input size (cf. Section 4.1). Even for the largest instance, it terminates in less than one second. The GDC+GAI algorithm, which runs GDC first and then GAI, also terminates within the time limit for all instances. In practice, GAI is much faster than GF and GAF because it may insert multiple arcs per iteration leading to fewer iterations overall. While the algorithm GAF+GM, on average, computes slightly better solutions than GDC+GAI, algorithm GDC+GAI is significantly faster on large instances. The average lower bound on the transitivity index computed by GDC+GAI is 0.61, i.e., the Google+ ego networks exhibit a slightly higher transitivity compared to the Twitter ego networks.

### 5.2 Twitter interaction network of the 117th US Congress

In this section we perform a qualitative analysis on a Twitter network of members of the 117th Congress of the United States. This network has been published by Fink et al. (2023) who have measured for each pair of Twitter accounts of Members of Congress the reaction likelihood, i.e., the fraction of times the first account has reacted to a contribution of the second account. The thusly obtained digraph consists of 475 nodes and 13,289 directed edges with non-zero likelihood. In contrast to the ego networks from the previous section, this network is not anonymized. This allows us to qualitatively interpret the results. We construct an instance of the preordering problem by defining arc values $c$ to be the reaction likelihood minus 0.01. With this, the arc values reward arcs from $i$ to $j$ to be part of the preorder if $i$ reacts on more than 0.01 of the contributions of $j$ and penalizes it otherwise. The value 0.01 is a subjective choice that was made in order to obtain a preorder that reveals some interesting structure. Values too large or too small lead to very sparse or very dense preorders, both of which offer little opportunity to draw conclusions.

For this instance of the preordering problem, the optimal solution value is $P(c) = 11.875$ and an optimal solution is depicted in Figure 9 in the appendix. It is found by our ILP algorithm in 4.9 s. The upper bound is $B(c) = 15.017$, which results in a transitivity index $T(c) = 0.791$. In the visualization of the preorder in Figure 9, nodes are arranged in layers and all arcs are directed from left to right. The further a node is to the left, the more the corresponding account reacts to others. The further a node is to the right, the more reactions the corresponding account gets from others. One can see that accounts react more to accounts from members of the same political party (Table 2 in the appendix). There are two layers with particularly

many nodes. This is expected as dicuts are transitive (cf. Section 4.1). However, there are many nodes not contained in the two large layers (i.e., the optimal solution is not close to a dicut). This is an indication that the interactions exhibit non-trivial transitivity. The node with the most incoming arcs (i.e., the account that gets the most reactions) is that of Kevin McCarthy (labeled with $a$ in Figure 9), the Republican minority leader in the House of Representatives of the 117th US Congress. There is a cluster of three accounts that is isolated from the rest (labeled with $b$ in Figure 9). These accounts belong to Tom Carper, Chris Coons and Lisa Blunt Rochester, who are the three Members of Congress from Delaware. The node labeled with $c$ in Figure 9 corresponds to the account of Bernie Sanders, who is a party independent member and thus expected to have fewer incoming arcs. Overall, the preorder exhibits few clusters containing more than one account. This is expected because a cluster indicates that accounts within the cluster react to one another and that all accounts in this cluster react to and receive reactions from the same accounts outside this cluster, which is rare.

If we require the preorder to be symmetric (i.e., an equivalence relation/clustering), the optimal value is 5.256. This value being much smaller than $P(c) = 11.875$ suggests that preorders are a better fit to this network than equivalence relations.

## 6 Conclusion

The NP-hard preordering problem is both a joint relaxation and a hybrid of the clique partitioning problem, from which the symmetry constraint is dropped, and the partial ordering problem, from which the antisymmetry constraint is dropped. A 4-approximation, computable in linear time, is given by the maximum dicut of the subgraph of all arcs with a positive value. Complementary to a local search algorithm by Böcker et al. (2009) that we call *greedy arc fixation*, heuristics that we call *greedy arc insertion* and *greedy moving* are defined analogously to algorithms known for clustering. The ILP formulation (1)–(3) of the preordering problem is tightened by odd closed walk inequalities that define facets of the preordering polytope.

In an application of the preordering problem and the algorithms described above to the task of jointly clustering and ordering the accounts of the social networks published by Fink et al. (2023) and Leskovec & Mcauley (2012), we observe qualitatively that preorders output by the algorithms differ from equivalence relations found by clustering, partial orders found by partial ordering, and preorders found by successive clustering and partial ordering. In addition, we observe quantitatively that high-quality solutions are found in practice by greedy arc fixation despite it not providing any approximation guarantee. Alternatively, first constructing a 4-approximation in the form of a dicut and then performing a local search by greedy arc insertion initialized with the dicut also finds high-quality solutions. Comparing upper and lower bounds, we observe that a substantial fraction of the LP gap is closed by adding odd closed walk inequalities.

**Broader Impact Statement**

This paper presents work whose goal is to advance the field of machine learning. The algorithms we introduce can be applied to social networks, including networks of individuals. They output relations that are transitive. The assumption that a social network is transitive simplifies and possibly contradicts reality. Working with this assumption is potentially controversial. In particular, the algorithms we define can be applied to identify the part of a relation on a social network that is not transitive. This can potentially serve as a basis for influencing a social network to become more or less transitive, which can have ethical implications.

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

# A  Appendix

## A.1  Discussion of reflexivity

The feasible solutions to the ILP formulation (1)–(3) of the preordering problem are precisely the characteristic vectors of the *reflexive reductions* of the preorders on $V$. There are two main reasons for our preference of this formulation over a version that would use the characteristic vectors of the preorders directly.

Firstly, we are interested in approximating solutions to the preordering problem. The quality of an approximation algorithm is usually defined as the maximum *multiplicative* error this algorithm commits over all instances of the problem. By assigning values also to the reflexive pairs and requiring a solution $A \subseteq V^2$ to be reflexive and transitive, the total value of every solution includes all values of transitive pairs. In particular, changing the values of the reflexive pairs merely shifts the values of all solutions by the same amount. Shifting the values of all solutions by a constant amount does not change the set of optimal solutions. However, it affects the ratio of values of solutions and thus how well one solution approximates another solution. Restricting our attention to the reflexive reductions of preorders is further justified by a lemma proven by Wakabayashi (1998) stating that the decision version of the problem is NP-complete for the reflexive version if and only if it is NP-complete for the irreflexive version.

Secondly, the convex hull of the characteristic vectors of transitive and reflexive relations is a polytope contained in a $|V_2|$-dimensional subspace of $\mathbb{R}^{V^2}$ (namely the subspace defined by the equalities $x_{ii} = 1$ for $i \in V$). By restricting the polytope to the space $\mathbb{R}^{V_2}$ (i.e., the convex hull of the feasible solutions to the ILP (1)–(3)), this complication is avoided.

## A.2  Derandomization of the max-dicut algorithm

It is well known that the randomized 4-approximation algorithm for max-dicut can be derandomized (Halperin & Zwick, 2001; Bar-Noy & Lampis, 2012). For completeness, we reproduce the derandomization due to Theorem 3 of Bar-Noy & Lampis (2012) below. Additionally, we show that this algorithm can be implemented to run in $\mathcal{O}(|V|^2)$.

Let $G = (V, E)$ be a directed graph with edge weights $w : E \to \mathbb{R}_{\geq 0}$. In the following, let $S, \bar{S} \subseteq V$ with $S \cap \bar{S} = \emptyset$. They define the partial dicut $\{ij \in E \mid i \in S, j \in \bar{S}\}$. We denote the expected value of the dicut that is obtained by randomly assigning elements in $V \setminus (S \cup \bar{S})$ to either $S$ or $\bar{S}$ by $E_{S\bar{S}}$. It holds that

$$
E_{S\bar{S}} = \sum_{\substack{ij \in E: \\ i \in S, j \in \bar{S}}} w_{ij} + \sum_{\substack{ij \in E: i \in S, \\ j \in V \setminus (S \cup \bar{S})}} \frac{w_{ij}}{2}
$$
$$
+ \sum_{\substack{ij \in E: j \in \bar{S}, \\ i \in V \setminus (S \cup \bar{S})}} \frac{w_{ij}}{2} + \sum_{\substack{ij \in E: \\ i,j \in V \setminus (S \cup \bar{S})}} \frac{w_{ij}}{4} \quad .
$$

For $k \in V \setminus (S \cup \bar{S})$, let $g_k$ denote the change in the expected value after assigning $k$ to $S$. A simple calculation yields

$$g_k = E_{S \cup \{k\} \bar{S}} - E_{S \bar{S}} = \sum_{\substack{j \in \bar{S}: \\ kj \in E}} \frac{w_{kj}}{2} - \sum_{\substack{i \in S: \\ ik \in E}} \frac{w_{ik}}{2}$$
$$+ \sum_{\substack{j \in V \setminus (S \cup \{k\} \cup \bar{S}): \\ kj \in E}} \frac{w_{kj}}{4} - \sum_{\substack{i \in V \setminus (S \cup \{k\} \cup \bar{S}): \\ ik \in E}} \frac{w_{ik}}{4}$$

and

$$E_{S \bar{S} \cup \{k\}} - E_{S \bar{S}} = -g_k \ .$$

If $g_k$ is non-negative, assigning $k$ to $S$ does not decrease the expected value; otherwise, assigning $k$ to $\bar{S}$ does not decrease the expected value.

When assigning all elements randomly, the expected value is $E_{\emptyset \emptyset} = \sum_{ij \in E} \frac{w_{ij}}{4}$. By iteratively assigning elements $k \in V \setminus (S \cup \bar{S})$ to $S$ or $\bar{S}$ based on the sign of $g_k$, a dicut is obtained with value at least $E_{\emptyset \emptyset}$ and, therefore, a 4-approximation to the max-dicut problem.

Algorithm 1 implements this algorithm with two additional improvements: Firstly, in each iteration, the element $k \in V \setminus (S \cup \bar{S})$ for which $|g_k|$ is greatest is selected. This maximizes the expected value after assigning $k$. Secondly, the values $g_k$ for $k \in V \setminus (S \cup \bar{S})$ are not computed from scratch in each iteration. Instead, they are computed once for $S = \bar{S} = \emptyset$, and after each iteration, these values are updated. With this, each iteration runs in time $\mathcal{O}(|V|)$ instead of $\mathcal{O}(|V|^2)$ that would be required to compute all $g$ values according to the formula above.

### A.3 Example instance

In this section we describe an instance of the preordering problem where the greedy arc fixation algorithm (cf. Section 4.2) does not achieve a 4-approximation. In contrast to Figure 4, on this instance no ties occur throughout the execution of the algorithm and thus no unfavorable tie-breaking. The arc values of such an instance are given by the matrix below.

$$\begin{pmatrix} 0 & -9976 & -20009 & -10060 & -20099 & 10033 \\ -9908 & 0 & 10025 & -19965 & 9996 & 6 \\ 10049 & -10048 & 0 & 10018 & -19971 & -20025 \\ -19943 & -9914 & -10076 & 0 & -19984 & 10014 \\ 9950 & -91 & -19988 & 10021 & 0 & -19934 \\ -20025 & 10086 & 9947 & -10032 & -10035 & 0 \end{pmatrix}$$

This matrix was generated by uniformly sampling values from $\{-2, -1, 0, 1, 2\}$ in order to obtain a difficult instance and then scaling these values by 10000 and adding uniformly sampled noise from $\{-100, \ldots, 100\}$ in order to obtain unique values. The optimal preorder has an objective value of 50130. The preorder found by the greedy arc fixation algorithm has an objective value 10280, yielding an approximation ratio of $4.88 > 4$. The greedy arc fixation algorithm performs the following sequence of fixations: $x_{\{5,4\}} = 0$, $x_{\{0,4\}} = 0$, $x_{\{0,1\}} = 0$, $x_{\{5,2\}} = 0$, $x_{\{3,1\}} = 0$, $x_{\{5,1\}} = 0$, $x_{\{0,2\}} = 0$, $x_{\{3,4\}} = 0$, $x_{\{3,2\}} = 0$, $x_{\{5,0\}} = 0$, $x_{\{5,3\}} = 0$, $x_{\{2,1\}} = 0$, $x_{\{2,5\}} = 1$, $x_{\{4,2\}} = 0$, $x_{\{1,3\}} = 1$, $x_{\{3,0\}} = 0$, $x_{\{1,5\}} = 1$, $x_{\{4,1\}} = 0$, $x_{\{1,0\}} = 1$, $x_{\{2,4\}} = 0$, $x_{\{4,5\}} = 1$, $x_{\{0,3\}} = 0$, $x_{\{2,0\}} = 1$, $x_{\{0,5\}} = 1$, $x_{\{1,2\}} = 1$, $x_{\{4,3\}} = 1$, $x_{\{2,3\}} = 1$, $x_{\{3,5\}} = 1$, $x_{\{1,4\}} = 1$, $x_{\{4,0\}} = 1$. Each fixation is the unique best fixation and no ties occur.

### A.4 Preorder of the 117th US Congress

| Party | Democrats | Republicans | Other |
|---|---|---|---|
| Democrats | 409 | 49 | 0 |
| Republicans | 53 | 531 | 0 |
| Other | 18 | 3 | 0 |

Table 2: The number in row A and column B is the number of arcs in the preorder depicted in Figure 9 from members of party A to party B.

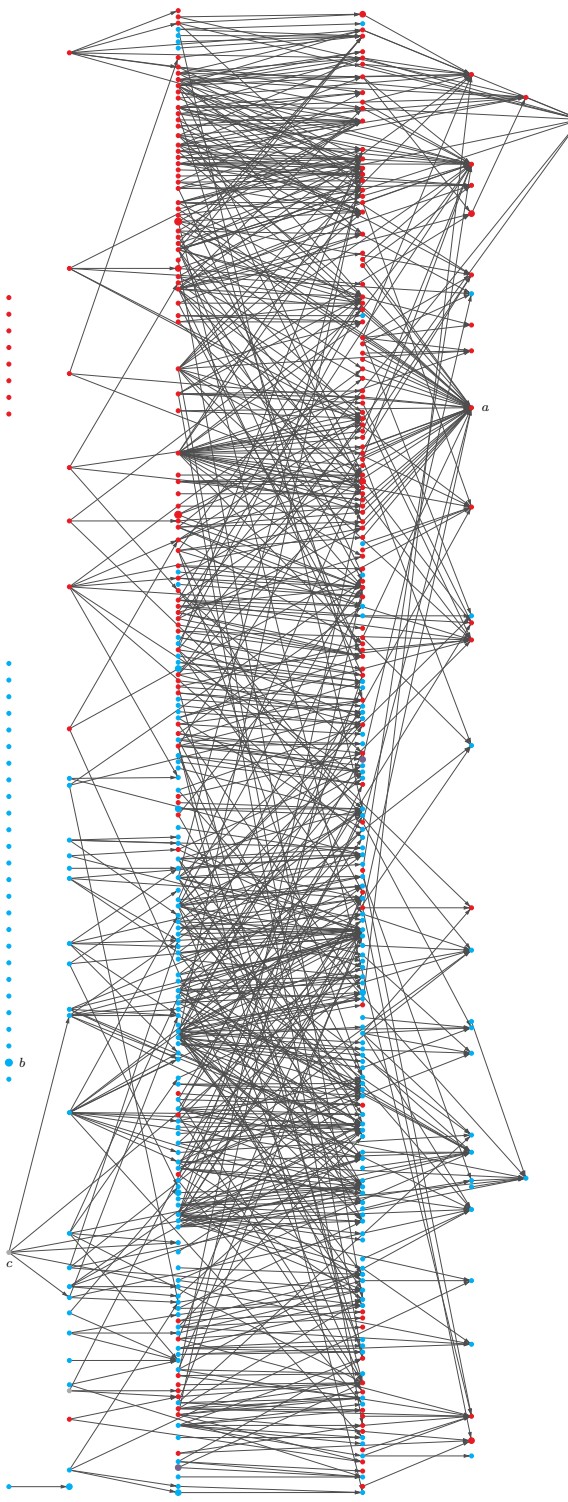

Figure 9: Preorder of Twitter accounts of members of the 117th US Congress. Each node corresponds to a cluster of accounts and the size of the node is proportional to the size of the cluster (most clusters contain just a single account.). The color of the node corresponds to the party membership (Democrats: blue, Republicans: red, Other: gray), where clusters with mixed membership have a mixed color. The depicted edges are those of the transitive reduction of the partial order on the clusterings. The layout is computed using the dot algorithm implemented in Graphviz (Ellson et al., 2004).

