# OpenReview forum: "Algorithms for the preordering problem and their application to the task of jointly clustering and ordering the accounts of a social network"
_TMLR — Accepted by TMLR_

### Review · Reviewer_D6Ey · 2025-09-07

**Summary Of Contributions:**

This paper studies algorithms for the maximum value preordering problem. The authors present (a) a simple randomization-based lower bound with a 4-approximation guarantee, (b) an LP-based relaxation strengthened with odd closed walk inequalities to provide tighter upper bounds, and (c) several heuristic approaches, including a previously published baseline and two new greedy variants. The paper includes the necessary proofs, such as the relationship between preorders and dicuts, and provides clear arguments for the approximation guarantees. Beyond the theoretical contributions, the paper offers detailed empirical evaluation on real-world social network data from Google+, Twitter ego networks, and the Twitter interaction network of members of the 117th U.S. Congress, comparing runtimes and solution quality across algorithms.

**Audience:**

Yes

**Audience Explanation:**

The paper systematically analyzes several algorithms for the maximum value preordering problem including a randomized algorithm for a lower bound and an approximate upper bound using linear programming, apart from 3 greedy algorithms including 2 proposed ones. These algorithms are described in detail with theoretical evidence where needed. Additionally, the paper provides empirical evidence about performance and runtime for each of the methods and several combinations of the methods as well. The depth of analysis and detailed explanation of the method makes it an interesting paper for TMLR's audience subject to the requested changes being made.

**Broader Impact Concerns:**

The broader impact statement is generally sufficient, but one point worth emphasizing is the risk of misinterpretation when applying these methods to political or social datasets. Enforcing preorder structures can exaggerate hierarchy, partisanship, or suggest influence chains that are not truly causal. It would help if the authors clearly state that preorders are a modeling assumption rather than ground-truth evidence about social interactions.

**Claims And Evidence:**

Yes

**Claims Explanation:**

The claims are supported by both theoretical proofs and empirical results. The paper provides clear arguments for the 4-approximation guarantee and for the relationship between preorders and dicuts, and it cites prior hardness results appropriately. The empirical evaluation on large social network datasets (Twitter, Google+, U.S. Congress) is detailed and shows that the heuristics perform well and that the LP relaxation with odd-closed walk inequalities improves the quality of the upper bound.

**Requested Changes:**

Changes required for acceptance:
- While the paper provides distribution plots (e.g., Figure 5), the results section would benefit from a clear summary table reporting average/median objective values, transitivity indices, and runtimes across all ego networks. This would make the empirical comparison between preorder, clustering, and partial order easier to interpret at a glance, and also support claims such as "The objective values are highest for the preordering problem and lowest for the clustering problem".
- In section 5.1.1, it would be good to quantify the amount of disagreement between the solutions of each method. The general messaging of the paper based on Figure 6, is that preorder has less disagreements than clustering and partial ordering, but concrete evidence would give more evidence to support this claim.
- In section 5.1.2, report quantitative metrics in addition to the results shown in Figure 7. The data in the plot does provide some evidence but the granularity of the figure and how close the different methods makes it challenging to directly interpret the data.
- In section 5.2, a) justify why 0.01 is a fair choice (eg. by analyzing mean/median proportion of interactions, 10th percentile, etc.) for penalizing arcs, b) quantify how often party members are connected to other members of their/other parties to support statements such as "one can see that accounts react more to accounts from members of the same political party", c) explain why it is reasonable for most clusters to only contain one account in the preorder

Minor changes:
- For consistency in metrics across the paper, it would be good to only report the performance normalized by |E| (eg. table 1)
- Define "reaction likelihood" in section 5.2

---

> ### Author Response · Authors · 2025-11-24
> **Response to Review**
>
> We thank the reviewer for their detailed review with constructive suggestions for improvement. Below we address all points one by one.
>
> - **Summary tables:** We have added tables summarizing quantitative metrics to Figures 5, 7 and 8. In Figure 8, we do not report median runtimes but instead fit polynomials and report their degree. We do this because the runtimes follow a clear trend here, and we believe that the polynomial degree is more meaningful than the median runtime.
>
> - **Quantifying disagreement in Section 5.1.1:** Given a solution with objective value $Obj$ and the number of edges $|E|$, the disagreement of that solution is $|E| - Obj$. Therefore, stating the number of disagreements alongside the objective values would be redundant. We have made this more clear by adding to the caption of Figure 6 the sentence: "The total number of disagreements is precisely the number of edges minus the objective value stated in Table 1".
>
> - **Quantitative metrics for Figure 7 in Section 5.1.2:** We agree with the observation that data in the right plot of Figure 7 is challenging to read. As stated above, we have added a table next to the plot listing the corresponding quantitative values.
>
> - **Section 5.2**
>     - Reasoning for 0.01: This value is not obtained by an analysis of the distribution of reaction likelihoods (mean/medians/percentiles). Instead, we have chosen this value because it leads to a preorder that reveals some interesting structure. Values too large or too small lead to very sparse or very dense preorders, both of which offer little opportunity to draw conclusions. That being said, the mean value of all non-zero reaction likelihoods (13289 out of all 225150 pairs have a non-zero reaction likelihood) is 0.00581. Using this value instead of 0.01 leads to very similar results. We will gladly use this value instead if the reviewer suggests it. Moreover, we will add the following sentences to the end of the first paragraph of Section 5.2: "The value 0.01 (0.00581) is a subjective choice that was made in order to obtain a preorder that reveals some interesting structure. Values too large or too small lead to very sparse or very dense preorders, both of which offer little opportunity to draw conclusions."
>     - We have added a confusion matrix that states how many arcs of the preorder connect members between specific parties (Table 2 in the appendix).
>     - To explain why it is reasonable that most clusters of the preorder contain only one account, we have added the following sentences the end of the second paragraph: "Overall, the preorder exhibits few clusters containing more than one account. This is expected because a cluster indicates that accounts within the cluster react to one another and that all accounts in this cluster react to and receive reactions from the same accounts outside this cluster, which is rare."
>
>
> - **Consistent metrics:** In Table 1, we prefer to report the objective value because it is easier to relate these numbers to the illustrations in Figure 6. For consistency, we have added the normalized values to this table.
>
> - To clarify the term **"reaction likelihood"** we have reformulated the second sentence of Section 5.2 as follows: "This network has been published by Fink et al. (2023) who have measured for each pair of Twitter accounts of Members of Congress the reaction likelihood, i.e. the fraction of times the first account has reacted to a contribution of the second account."

---

### Review · Reviewer_Msff · 2025-10-03

**Summary Of Contributions:**

This paper proposed algorithms to resolve the preordering problem. The paper detailed the proposed algorithms and evaluated them on the ego networks of Twitter and Google+. The results show the algorithms could solve the problem effectively.

Strengths
1. The algorithms are well-written, and the experiments on the proposed algorithms are comprehensive.
2. The ablation study of the combination of algorithms are good.

Weaknesses

1. There is no comparison to others' work; how do the algorithms compare to the state-of-the-art methods?
2. It is hard to connect the paper to the applications, e.g. in the social network, why solving preordering problems is important, what is the consequence if the accuracy is not high enough. Those connections could further strengthen the paper.

**Audience:**

Yes

**Audience Explanation:**

Yes, solving social network problem is important in the machine learning, especially large-scale network and has to solve it with a limited time.

**Claims And Evidence:**

Yes

**Claims Explanation:**

N/A

**Requested Changes:**

1. See weaknesses.  Especially, the comparison to other works.
2. Is it possible to evaluate the algorithms' performance without time constraints? My concern is that is it possible one algorithm just can not find a good solution in the short time but it could lead to a better solution later.

---

> ### Author Response · Authors · 2025-11-24
> **Response to Review**
>
> We thank the reviewer for their insightful feedback. Below we address all points one by one.
>
> - **Comparison to the state-of-the-art methods**:
> To the best of our knowledge, Jacob et al. (2008) and Böcker et al. (2009) are the only publications addressing practical algorithms for this problem. Unfortunately, neither the source code nor the problem instances are publicly available. Therefore, we have synthesized instances as described by Böcker et al. (2009) and have solved these with our ILP solver. The largest instances (500 nodes) are solved in 7 to 14 seconds by our implementation, while Böcker et al. (2009) report runtimes of 100 to 1400 seconds (see Figure 3 of Böcker et al. (2009)) for all their algorithms. We cannot determine whether the faster runtimes of our algorithm are due to better hardware, an improved implementation of the ILP solver, or other reasons. Hence, we prefer not to include these results in our article.
>
> - **In the social network, why solving preordering problems is important?**:
> While we are no sociologists, we speculate that understanding the transitivity in social networks can lead to valuable insights. We believe that our rigorous approach to quantifying transitivity with the transitivity index will be a useful tool for sociologists. Moreover, to better convey the intuition, we have added the following sentences to the end of the last paragraph of Section 3:
> "In the context of social networks, the transitivity index can be interpreted as a measure for how homogeneous the information flow is. If the transitivity index is high, the information flow is homogeneous: If $a$ receives information from $b$ and $b$ receives information from $c$, the $a$ also receives information form $c$. If the index is low, the flow of information is less structured."
>
> - **What is the consequence if the accuracy is not high enough?**:
> Firstly, if the computed solution is far from optimal, it is difficult to judge whether the preordering problem is the wrong model for the task at hand (modeling error) or the right model with a solution that is just not good enough (optimization error).
> Secondly, for a practical example, suppose the solution to the preordering problem is used to streamline the information flow between coworkers of a company. Making changes to the company structure (e.g. relocating employees) can be costly. Therefore, it is desirable that the solution to the preordering problem is as good as possible.
>
> - **Is it possible to evaluate the algorithms' performance without time constraints?**:
> The imposed time limit for the ILP solver is necessary because its worst case time complexity is exponential in the input size. The main bottleneck is that if the LP bound is to weak, the search tree of the branch and bound algorithm within the ILP solver becomes too large. If this happens, progress of the ILP solver becomes very slow. In our experiments, 459 instances of the preordering problem are solved to optimality within 100 seconds, 493 instances are solved to optimality within 400 seconds, and 499 instances are solved to optimality within out set time limit of 500 seconds. This illustrates that increasing the time limit beyond 500 seconds is unlikely to result in many more solved instances.
> The situation is very similar for the ILP solver for the clustering and partial ordering problems.
> The analysis in Figure 5 is restricted to those instances for which all algorithms have terminated within the time limit.
> Therefore, these results are not affected by some algorithms not finding a good solution within the time limit.

---

### Review · Reviewer_1iLa · 2025-11-18

**Summary Of Contributions:**

This paper introduced and implemented algorithms for the preordering problem.

Strengths:
- In-depth with proof

Weaknesses:
- Crowded figures
- No comparison of all algorithms together

**Audience:**

Yes

**Audience Explanation:**

This paper shows the efficiency of the algorithms introduced, which can lead to further discussion.

**Broader Impact Concerns:**

There are no broader impact concerns.

**Claims And Evidence:**

Yes

**Claims Explanation:**

Each theorem has a proof and there are empirical results.

**Requested Changes:**

- Figures 6 and 9 are cluttered, maybe move them to appendix
- Maybe include a table with the number of optimal solutions found for all algorithms (5.1.2)
- Labels on figure 6

---

> ### Author Response · Authors · 2025-11-24
> **Response to Review**
>
> We thank the reviewer for the suggestions for improvement of the article. Below we address all points one by one.
>
>
> - **No comparison of all algorithms together:**
> The boxplot in Figure 7 depicts the difference in the objective value of all six heuristics to the optimal objective value computed by the ILP solver. In this way, all algorithms designed for computing feasible solutions to the preordering problem (ILP solver and six heuristics) are compared in this figure.
> The ILP solvers for the other problems (clustering and partial ordering) are not contained in this figure because they are not directly designed to compute good feasible solutions to the preordering problem.
> The LP solver is not contained in this figure because it does not compute feasible solutions and instead computes only upper bounds.
>
> Figure 8 depicts the runtimes of all heuristic algorithms for the preordering problem but not that of the ILP solver. This is because the ILP solver is impractical for instances of this size.
>
>
> - **Figures 6 and 9 are cluttered, maybe move them to appendix:**
> We have de-cluttered Figure 6 by inserting separating lines and network IDs that distinguish the individual instances much more clearly. We have also followed your suggestion and have moved Figure 9 to the appendix.
>
> - **Include a table with the number of optimal solutions found for all algorithms (5.1.2):**
> We have included in Figure 7 a table with the number of instances for which heuristic algorithms find an optimal solution.
>
>
> - **Labels on figure 6:**
> We have added the instance IDs as labels to each row.

---

### Decision · Action_Editor_jygn · 2026-01-02

**Recommendation:** Accept as is

**Audience:**

Yes

**Audience Explanation:**

While this paper is somewhat on the fringe of the TMLR audience, I think that the paper addresses a problem that would be of interest to some in the community.  The clustering problem, for instance, is a core problem in machine learning, and this paper falls within that general area.

**Claims And Evidence:**

Yes

**Claims Explanation:**

The paper presents several algorithms for the preordering problem, including some with theoretical guarantees.  These guarantees appear to be correct, and to build upon previous work such as Boecker et al.  From a theoretical and algorithmic perspective, the paper has sufficient evidence.

One of the concerns given by reviewers was due to the limited empirical evaluation.  I agree that there is somewhat limited empirical evaluation in the paper, but the authors do compare against the greedy arc fixation algorithm of Boecker et al.  The authors mention the algorithm of Jacob et al, but this seems to be a heuristically-driven algorithm as opposed to the more theoretically-driven algorithms in this paper.  Given the relative lack of existing algorithms here, I think the empirical justification given is sufficient.  In particular, on small instances, the authors can compare against the optimal algorithms that are computing via integer linear programming, along with several variants of the methods that they develop.